# Malnutrition in Older Adults—Effect on Falls and Fractures: A Narrative Review

**DOI:** 10.3390/nu14153123

**Published:** 2022-07-29

**Authors:** Malgorzata Kupisz-Urbanska, Ewa Marcinowska-Suchowierska

**Affiliations:** 1Department of Geriatrics, Centre of Postgraduate Medical Education, 02-673 Warsaw, Poland; 2Department of Internal Medicine and Geriatric Cardiology, Centre of Postgraduate Medical Education, 02-673 Warsaw, Poland; emarcinowska1@gmail.com; 3Department of Geriatrics and Gerontology, School of Public Health, Medical Centre of Postgraduate Education, 02-673 Warsaw, Poland

**Keywords:** older adults, malnutrition, fracture risk, falls, immobility, functional decline

## Abstract

Malnutrition in older adults impacts health status, increased mortality, and morbidity. Malnutrition may increase the development of geriatric syndromes and contribute to a higher prevalence of falls and osteoporotic fractures that lead to loss of independence and an increased rate of institutionalization. The role of malnutrition in the pathogenesis of other geriatric syndromes seems to be well established. However, the data concerning nutritional interventions are confounding. Moreover, long-term undernutrition seems to be one of the factors that strongly influences the efficacy of interventions. This review outlines the current literature on this topic, and aims to guide physicians to make proper decisions to prevent the vicious cycle of falls, fractures, and their negative outcomes in patients with malnutrition.

## 1. Introduction

An older adult is typically described as an individual aged 65 years or older. Among this group, persons aged 75 years or older are usually differentiated, as they more often represent a specific profile of “geriatric patient” characterized by a higher degree of frailty, functional decline, and multiple active diseases [1]. The geriatric patient has a greater need for omnidirectional care (including diagnosis, adjusted treatment, rehabilitation, and psychological and social care) and requires a holistic approach, usually with a geriatric comprehensive assessment (GCA) to avoid a loss of independence. Nutritional status in older adults reflects the health status and seems to play a vital role in the musculoskeletal system, bone density and strength, fall risk, postural instability, and immobility, as well as many other negative outcomes in different systems and organs. The population of oldest-old patients is still growing, and they represent a group in which aging processes that are accumulated due to the influence of malnutrition and/or malnourishment on falls could be higher than in younger groups. Several studies suggest that malnutrition in oldest-old patients is one of the factors that should be considered when assessing risk fracture.

### 1.1. Aim

The aim of this narrative review was to estimate the problem of malnutrition and the risk of malnutrition in older adults and assess the influence of malnutrition on the risk of falls and fractures in clinical practice.

### 1.2. Malnutrition in Older Adults

Three specific features characterize geriatric syndromes: the prevalence of malnutrition is higher in this age group; the reasons for malnutrition are multifactorial; and health outcomes are more serious than in younger individuals. Malnutrition is one of the geriatric syndromes that greatly reduces regenerative and adaptive capacity, and remains an an important modulator of successful aging.

### 1.3. Diagnosis, Epidemiology, Estimation

The European Society for Parenteral and Enteral Nutrition (ESPEN) guidelines on clinical nutrition in geriatrics recommends using a combination of two features to diagnose malnutrition in oldest-old patients. The first is the phenotype criterion (such as weight loss being nonvolitional, reduced muscle mass, or low body mass index (BMI)) and the second one is the etiologic criterion (such as severe diseases, especially when coexisting with inflammation, as well as reduced food intake and/or malabsorption). Experts also recommend diagnosing older patients who are at risk of malnutrition. The global consensus approach recommends diagnosing patients with higher malnutrition risk when oral intake is significantly reduced (≥50% of requirements for more than three days), or when risk factors that could reduce dietary intake or increase requirements (such us dementia, depression, acute disease, mobility reduction, or chewing or swallowing problems) are present [2]. The prevalence of malnutrition increases with functional decline and general health status deterioration. This contributes greatly to a vicious cycle, as one of typical negative outcomes of malnutrition is the further deterioration of health status. The reported prevalence rates strongly depend on the definition applied. Prevalence of undernutrition differs from about 10% in independently living older persons to about 65% in hospitalized patients [3,4]. The Global Leadership Initiative on Malnutrition (GLIM) underscored the key role in a graduated diagnostic scheme: the first step of diagnosing is screening to identify if the patient is “at risk”, the second step is the assessment of undernutrition, the third step is diagnosing, and the final step is an evaluation of severity [5,6].

Undernutrition in older people can be estimated using different tools, such as scales, anthropometric measurements, body composition parameters, and laboratory blood tests, as well as formulas such as the geriatric nutritional risk index (GNRI), which connects some laboratory tests and anthropometric features. Undoubtedly, in clinical research and in clinical practice, the first step should be the assessment of malnutrition risk, followed by the diagnosis of malnutrition. However, the two groups of geriatric patients should be given a regular, detailed estimation of malnutrition risk and diagnosis of malnutrition along with an assessment of other geriatric syndromes. Such other syndromes, such as functional decline, are crucial risk factors for malnutrition. Moreover, in the geriatric population, it is challenging to distinguish one factor that contributes to malnutrition (including reduced intake or assimilation of nutrients, disease-associated mechanisms, injury, or inflammation) and multifactorial mechanisms that are most common [7].

*Scales* used for the screening and evaluation of malnutrition in oldest-old patients in clinical practice include the Mini Nutritional Assessment Short Form and the Mini Nutritional Assessment Full Form (MNA-SF and MNA-FF, respectively), which are the most common scales in geriatric comprehensive assessment. Other available scales include the MST—Malnutrition Screening Tool, GNS—Geriatric Nutritional Status, and CONUT—Controlling Nutritional Status. However, the current ESPEN recommendations highlight that, beyond choosing the adjusted scale, the implementation of interventions based on effective tools plays a crucial role. Experts encourage clinicians to employ not only screening or evaluation, but also to apply interventions to improve clinical outcomes.

*Anthropometric measurements* greatly contribute to the malnutrition assessment. Body mass index (BMI) is defined as the ratio of body mass (kg)/height (cm)^2^. The relationship between BMI and clinical outcomes has been recently discussed. It is probable that in older individuals, BMI alone is not enough to define malnutrition, and this phenomenon is more obvious with age. Some authors suggested the waist/hip ratio (WHR) was better, as it was more specific for abdominal obesity, whereas body mass index more reflects the lean body mass (LBM). However, WHR or waist (recommended now) seems not to be suitable for older adults, as the fluctuations strongly depend on body composition changes coexisting with musculoskeletal changes. The cutoff for BMI for adults is between 18.5 kg/m^2^ and 24.9 kg/m^2^, but for older adults, it should be greater—between 24 kg/m^2^ and 27 kg/m^2^ [8]. Even though anthropometric variables can be misleading, they are an easily applied method that can be an initial assessment for every geriatric patient.

Body composition (BC) estimation methods include measurements such as lean body mass (LBM), fat mass (FM), body fat, and percentage body fat (BF%). Body composition can be measured using computer tomography (CT), bioimpedance (BI), or dual-energy X-ray absorptiometry (DEXA)—a recommended method to assess BC. Assessing body composition in oldest-old patients has greater importance, especially when considering clinical outcomes, particularly the increased risk of sarcopenia and disability. Reduced muscle mass was one of the top five ranked criteria for diagnosing malnutrition recommended by GLIM. The other criteria were nonvolitional weight loss, low BMI, reduced food intake or assimilation, and disease burden or inflammation. In the published data, body composition remains more adequate for malnutrition phenotype assessment in older adults. Loss of skeletal muscle mass (SMM) is a physiological condition manifested by aging, a diminished whole-body metabolic rate, and a reduction in oxidative capacities. This contributes to lower energy expenditure from basal metabolism, which may lead to weight gain that is mostly fat mass. In succession, the accumulation of intramyocellular lipids promotes lipotoxicity and amplifies age-associated inflammation, mitochondrial dysfunction, and other metabolic changes such as insulin resistance (IR). Hormonal dysregulation, proinflammatory cytokines, myokines (including insulin-like growth factor 1 (IGF-1), myostatin, and irisin), and adipokines (including adiponectin and resistin) constitute crucial factors in muscle homeostasis and the pathogenesis of sarcopenic obesity. Furthermore, it is important to emphasize that a vicious cycle may occur in which a negative interaction is maintained not only between muscle and adipose tissue, but also with bone.

Moreover, aging-related changes in body composition can affect the validity of the methods mentioned above. Regardless of the body mass index approach, the predicted BF% increases with age in both males and females. The appropriate contribution of FM and LBM to body weight is an essential risk indicator of major issues concerning geriatric care [9]. The range of norm of BF% in adults differs greatly between trained/untrained men (2–24%) and trained/untrained women (10–31%). In oldest-old patients, the percentage body fat is greater and excess 40%, however the range of the norm depends on age and sex, and is still an issue of a debate [10].

*Laboratory tests* can be useful in assessing qualitative and quantitative malnutrition. The most useful parameters available in everyday practice include serum albumin, prealbumin, total protein, transferrin, lymphocyte count, and level of total cholesterol, as well as vitamin D, acidic folium, and vitamin B12 serum concentration. The range of laboratory tests to assess micronutrient abnormalities and laboratory indicators of malnutrition for older adults have the same cutoffs as for adults. Among others, the evaluation of chosen laboratory tests, in addition to the chosen geriatric scale, has become an effective tool to predict postoperative functional status [11]. However, it should be considered that there is no other range norm for serum plasma concentration for young adults, older adults, and oldest-old patients. For this reason, clinical manifestation and other geriatric issues should be considered in the diagnosis; while laboratory tests remain very helpful, they are additional in malnutrition risk and malnutrition diagnosis. These parameters also are indispensable in evaluation during comprehensive geriatric assessment.

Undernutrition and/or malnutrition in the geriatric population is a challenging health concern related not only to increased morbidity and mortality, but also physical decline, which has wide range of implications for mobility, instrumental activities of daily living, and general quality of life. Malnutrition is common and may contribute to the development of geriatric syndromes. In older adults, it can be seen as either involuntary weight loss or, in some cases, a low BMI; however, hidden deficiencies, especially micronutrient deficiencies, are more difficult to assess, and they are commonly overlooked in clinical practice. In developed countries, the most common source of malnutrition is disease, while acute chronic disorders have a high potential to result in or intensify undernutrition. Thus, as aging processes are risk factors for developing malnutrition and strongly contribute to disease development; older adults are the highest-risk population for nutritional risk and/or malnutrition. Undoubtedly, the etiology of malnutrition is complex and multifactorial, but the exacerbation of this process is facilitated by aging processes. Our comprehensive narrative review summarized the current evidence on the determinants and assessment of malnutrition in old adults while considering the clinical perspective and the contribution of malnutrition to falls and fractures. It is challenging to understand, identify, and treat malnutrition when falls and osteoporotic fractures occur. In some cases, the treatment target may include supplementation of macro- and/or micronutrients when diet alone is not sufficient to meet age-specific requirements [12]. The geriatric population remains at risk of protein-energy malnutrition (PEM). Considering PEM’s impact on health in older adults, including cognitive and physical decline, as well as a decline in the quality of life, the Healthy Diet for a Healthy Life’s joint programming initiative, Malnutrition in the Elderly (MaNuEL), is a knowledge hub focused on malnutrition. A meta-analysis based on six longitudinal studies from MaNuEL partner countries identified the determinants of malnutrition in older adults. The authors concluded that the key factors in predicting malnutrition were: increasing age, marital status (unmarried, separated, or divorced were at higher risk of malnutrition in comparison to married but not widowed), difficulties in walking 100 m and/or climbing stairs, and hospitalization. In women, cognitive impairment and receiving social support predicted malnutrition, as well as falling in the previous 2 years, hospitalization in the past year, and self-reported difficulties in climbing stairs when compared to men. Incorporating these findings into public health policy and clinical practice could support the early identification and management of malnutrition [13]. Based on the GLIM diagnostic scheme for screening, assessment, diagnosis, and grading of malnutrition and the available literature, the key points of the tailored process for geriatric comprehensive care are presented in Table 1.

## 2. Falls and Fractures—Consequences of Malnutrition

A wide range of consequences of malnutrition have a strong impact on the musculoskeletal system. The key role of undernutrition in the progress of sarcopenia is well established. Nevertheless, the correlation of malnutrition mechanisms influencing falls, fractures, and their outcomes is still an issue of debate, especially considering that observational and interventional studies have become more complex. The factors that play a crucial role in falls and fractures are long-term undernutrition, number of comorbidities, advanced age, and method of malnutrition assessment. 

### 2.1. Falls

The role of falls in health status is well established in the geriatric population. as they strongly affect losses in mobility and functional decline. Simultaneously, falls remain a key factor in osteoporotic fractures. The ratio of fractures is 7–9 times higher in older women with falls coexisting with osteoporosis or osteopenia, in comparison to women who only experienced falls or osteoporosis/osteopenia [14]. For this reason, fall prevention has become a goal of osteoporosis treatment. Malnutrition prevention is an important goal for fall prevention, especially when taking into consideration the effect of general muscle mass, strength, and function loss. Gusdal and colleagues conducted a cross-sectional registry study with 5919 older adults over 65 years old who were residents of nursing homes or dementia care units. They found that 77% of the subjects were at risk of falls, and nearly 60% were at risk of undernutrition. The authors emphasized that the most prevalent risk factors for falls were previous falls and cognitive decline; and for undernutrition, mild or severe dementia and/or depression. The findings of this study showed that the risk of falls and the risk of undernutrition were positively correlated [15]. In addition, a preliminary study assessing the relationship between muscle strength and protein intake in patients with hip fractures was published in 2020. The authors concluded that in patients with this type of osteoporotic fracture, insufficient protein intake was common and significantly correlated with lower muscle strength, which contributes strongly to falls [16]. These data were consistent with a review that summarized the impact of malnutrition and loss in muscle strength and function on clinical outcomes [17]. The authors of this review stated that nutritional status is a crucial factor for muscle strength and function, and malnutrition contributes strongly to functional decline. Lackoff and colleagues concluded that malnourished older adults experience harmful falls more often. Malnutrition and lower BMI were significant predictors of risk (*p* < 0.0001 and *p* = 0.26, respectively). The risk of harmful falls among malnourished patients were 8 times higher compared to the control group (without malnutrition). The authors suggested that malnutrition assessment should be included in fall risk assessment to identify the high-risk group [18]. Other researchers made similar observations. Poor nutritional status at the time of admission to the hospital is likely to be a predictor of in-hospital falls [19]. Despite of the wide range of knowledge concerning fall risk factors, a Swedish study with a sample of 5427 older adults showed that preventive interventions did not sufficiently follow current evidence. The authors assumed that despite the fact that prevalence ratio of falls was very high (79% for the study group), the planned interventions ratio was low, and often did not correspond with the risk factors [20].

### 2.2. Fracture Risk 

Loss of bone mass constitutes one of the critical characteristics of osteoporosis. Simultaneously, it was reported that aging increases fat mass and decreases muscle mass in both men and women [21]. A higher lean body mass was independently associated with a lower risk of vertebral fractures. Additionally, a greater waist circumference, which correlates with abdominal obesity, was independently associated with a higher risk of these fractures. These findings suggested that fat distribution may have a significant impact on the risk of vertebral fractures, central/abdominal obesity should be avoided, and muscle mass should be maintained [22]. An interesting approach to assessing the relationship between body composition and bone strength was presented by Leslie and colleagues [23]. The authors analyzed the body compositions of older adults during the DXA of the hip and spine, and based on developed mathematical formulas that calculated, among others, the total body fat, LBM, and bone strength indicators (SI), they found that there was no evidence that LBM and fat mass, as well as the SI of the femur, could predict major and hip fractures (FRAX values with or without BMD) in males and females. A higher fat mass was not independently associated with a higher FR over 5 years of follow-up. However, the authors of the same study applied the same methodology and found that the loss of total LBM was statistically greater in people with new fractures in the hip and major locations, while the loss of fat mass was significantly greater only in patients with new fractures of the femoral neck [24]. In addition, a 1SD loss of LBM was associated with a 10–13% higher risk of major fractures and a 29–38% increased risk of hip fracture (corrected for fat loss and other covariates). In the geriatric population, it is probable that lean body mass is only one factor that should be considered, and muscle strength and function could be distinctive characteristics for FR risk estimation.

Johansson and colleagues conducted a meta-analysis that focused on the relationship between fracture risk and BMI in women. They found that there was a significant association between body mass index and FR for all osteoporotic fractures, as well as for hip fractures. However, the authors stressed that this association was observed in the range of BMI between the lowest values in the study population up to 25 kg/m^2^, which constitutes the borderlines values for BMI for oldest-old patients. This association also was observed when adjusted with BMD, but the correlation was weaker [25]. A study was published a few years later for a female and male Korean population. With nearly 10 years follow-up observation, this study showed that a low lean mass and/or high percentage of body fat were related to a high fragility fracture risk in men, but not in women [26]. Nevertheless, only 4 of the 25 studies included in the meta-analysis of Johansson and colleagues concerned a group of oldest-old patients, while the Korean study concerned a group of patients younger than 60 years old. Maezava and colleagues compared the nutritional status (estimated via GNRI), level of hemoglobin, and renal function as measured by the estimated glomerular filtration rate (eGFR) between older women with femoral neck fracture (FNF) and osteoarthritis of the hip joint. They concluded that more than 40% of the patients before admission to the hospital had moderate or major risk estimated via GNRI, and found that FNF patients aged 75 and above had a significantly lower hemoglobin concentration and GNRI scores compared to hip osteoarthritis [27].

### 2.3. Negative Outcomes after Hip Fracture

As the population ages, there is an increase in the number of patients who require treatment for age-related conditions, such as surgical treatment after femoral neck or trochanteric fractures. Such operations are indispensable for remaining physically active. Malnutrition is a problem with multifaceted consequences, and among them, a higher number of hospitalization negative outcomes in comparison to patients without malnutrition [28]. The authors of the original study published in 2021 stressed that malnutrition, diagnosed via the GNRI, is not only a risk factor for osteoporotic vertebral compression fractures, but also greatly contributes to losses in functional decline (as assessed by activities of daily living (ADL)) and postoperative falls. Mariconda and colleagues stressed that a lower concentration of hemoglobin was one of the determinants of mortality after surgical treatment of hip fractures, especially concerning one-year survival in older old patients [29]. In addition, Sim and colleagues concluded that the preoperative nutritional status (including anemia and malnutrition features) was associated with morbidity greater than 30 days and mortality after operation and physical function decline [30]. Han and colleagues found that malnourished patients had longer hospital stays after operative treatment of hip fractures, a higher prevalence of pressure ulcers, higher in-patient mortality, and a nearly 3 times higher discharge to resident/nursing care. The authors stated that further research on nutritional support should be conducted to prevent hip fractures and reduce the number of complications. They also concluded that the prevalence of malnutrition and its negative outcomes increased with age, and varied from 6.7% among the 60–70-year age group to 24% in oldest-old patients (≥80 years of age) and nearly 30% in the oldest olds (≥90 years of age) [31]. In another study, researchers highlighted the role of coexisting geriatric syndromes, including cognitive impairment and delirium among older adults hospitalized with hip fractures. They assumed a strong correlation between delirium, cognitive impairment, and malnutrition [32].

## 3. Directions for Clinical Practice

Some authors aimed to identify effective interventions for malnutrition. It remains probable that skipping the first step—screening for malnutrition—is one of the key problems in undernutrition. Therefore, finding cases that assess the severity and planning intervention based on nutrition and exercises is indispensable, not only for hip fracture patients, but in all oldest-old patients with malnutrition to avoid lean mass loss. For patients admitted to the hospital after a hip fracture, screening for undernutrition is likely to be the first step in facilitating a fast nutritional intervention. Despite the wide range of observational studies confirming a strong relationship between malnutrition falls and fractures, the data concerning nutritional support are inconsistent. Some older double-blind, placebo-controlled randomized trials did not find any profit or found little benefit, but nutritional interventions in these studies was based on relatively low protein and calcium intake, and low compliance also was considered [33,34]. These RCTs did not find any benefit of the nutritional intervention on length of stay in hospital or any secondary clinical and functional endpoints (such as postoperative complications, functional status, quality of life, subsequent fractures, or mortality) [35]. It is likely that advanced process of aging, long-term undernutrition, the number of coexisting geriatric syndromes, comorbidities, medications, and social factors could confound the results. Undoubtedly, in clinical practice, more than one parameter of malnutrition should be considered when undernutrition is diagnosed. In oldest-old patients, who represent advanced aging processes, screening and prevention should be undertaken due to the high risk of undernutrition among this group of patients. From the clinical point of view, not only caloric and protein intake should be implemented, but also vitamin D supplementation, according to the recommendations for skeletal and extraskeletal effects.

Experts recommend a vitamin D supplementation dose of 800 to 2000 international units (IU) per day for adults who want to ensure a sufficient vitamin D status. These doses are also recommended for the treatment of vitamin D deficiency, but higher vitamin D doses (e.g., 6000 IU per day) may be used for the first 4 to 12 weeks of treatment if a rapid correction of vitamin D deficiency is clinically indicated before continuing with a maintenance dose of 800 to 2000 IU per day. Treatment success may be evaluated after 6 to 12 weeks in certain risk groups (e.g., patients with malabsorption syndromes) via measurement of serum 25(OH)D, with an aim to target concentrations of 30 to 50 ng/mL (75 to 125 nmol/L) [36,37].

A systematic review that focused on fall prevention in oldest-old patients based on recommendations published in the last three decades identified 15 high-quality practice guidelines for fall prevention and intervention [38]. The recommendations focused on risk assessment, prophylaxis, and management of falls in older adults. Experts recommended applying screening tools such as questionnaires and further gait and balance assessment. Multifactorial approaches were strongly recommended, including active management of fractures and osteoporosis as key elements in the prevention of falls. Medication review, exercise interventions, and environment modifications were also suggested. Recommendations strictly concerning malnutrition were focused on vitamin D supplementation, which is still under discussion. The strength of guidance varied greatly from strong to weak. This probably reflected the inconsistent data on vitamin D supplementation for fall prevention, especially in different study groups such as community, residential, or nursing-home care residents [39,40,41]. The authors of a recently published systematic review and meta-analysis focused on the effectiveness of multifactorial interventions (MIs) in preventing falls. They assumed that MIs with an exercise component have significantly reduced fall rates, and environmental modifications also had a significant influence. It was also highlighted that MIs significantly reduced fall rates in a high-risk subgroup and in healthy older adults when compared to usual care. However, no significant difference was revealed in the frail patient subgroup. Moreover, only a few of the studies included multifactorial intervention nutrition advice [42].

Energy intake in aggressive nutrition therapy (ANT) is defined as total energy expenditure (TEE) plus the amount of energy accumulated (EA). The application and effect of ANT differs depending on the etiology and characteristics of malnutrition, and is performed in malnourished and/or sarcopenic individuals to increase body weight and muscle mass. Determinants such as advanced age, precachexia, and mild and moderate dementia are indications for aggressive nutrition therapy. Nevertheless, ANT is contraindicated when acute disease or injury with severe inflammation, severe dementia, and reduced activity occurs, and usually should be combined with aggressive exercise (including resistance training), which may constitute a real obstacle in clinical practice [43]. Moreover, leucine, b-hydroxy-b-methylbutyrate (HMB), and protein have been shown to increase muscle mass. Leucine and HMB supplementation increase the activity of mammalian target of rapamycin complex 1 (mTORC1) and promote protein synthesis. This contributes to muscle mass increase in older adults, and was recommended for the geriatric frail population in an umbrella review of systematic reviews and meta-analyses. Still, the recommendations for protein supplementation were weaker than for leucine [44,45].

Concomitantly, ESPEN guidelines on clinical nutrition state that in older patients, malnutrition screening is necessary, and the recommended value for energy intake is 30 kcal per kg per day with at least 1 g of protein per kg body weight/day. Moreover, nutritional interventions should constitute a part of a multimodal and multidisciplinary intervention along with maintaining or increasing body weight to improve functional status and clinical outcomes. To achieve this aim, experts recommend, in older adults with malnutrition or who are at risk of malnutrition, fortified food and/or oral nutritional supplements that will provide 400 kcal/day or more, including at least 30 g of protein/day. The authors of these guidelines strongly recommended postoperative nutritional supplements in patients with hip fractures to reduce the risk of complications. A summary of the recommendations is presented in Figure 1.

Nevertheless, some limitations of the accessible publications should be considered. The main limitation was the age of studied groups: usually they represented the younger geriatric population, which is not representative of older adults, and especially oldest-old patients. Moreover, other coexisting geriatric syndromes were not screened, and there was a wide variation in the methodologies of the programs used (including malnutrition diagnosis, type of intervention, and period of observational or interventional studies).

## 4. Conclusions

Early detection of undernutrition, risk of falling, and fracture risk facilitate performing planned interventions, and is crucial in the geriatric population to prevent negative outcomes, functional decline, and institutionalization. Moreover, malnutrition contributes strongly to more harmful falls. Furthermore, patients with cognitive decline constitute the high-risk group for malnutrition and its complications. Hence, the systematic and individualized estimation of these factors can guide decisions regarding adjusted interventions. However, some studies showed that older adults estimated to be at risk of falling and/or malnutrition did not receive adequate interventions, and there are confounding data between assessed risk factors and planned and performed interventions [46]. According to the available literature, multifactorial interventions, including nutritional intervention, vitamin D supplementation, physical training, and treatment of underlying causes, has the best impact on health status in older adults. However, a recent meta-analysis suggested that vitamin D supplementation showed no significant reduction in fractures and falls, while older RCT meta-analyses revealed such a relationship. A Central and Eastern European expert consensus statement addressing the uncertainties regarding vitamin D, in the context of osteoporosis, is strongly recommended to ensure a sufficient vitamin D status in patients with an increased risk of fractures and falls. Recently published randomized control trials revealed that increasing calcium and protein intake via milk, yogurt, and cheese significantly reduced the risk of falls and fractures in aged care residents [47,48].

Undoubtedly, further studies are needed, especially focused on screening for the risk of malnutrition and malnutrition diagnosis in the older old population with musculoskeletal diseases.

## Figures and Tables

**Figure 1 nutrients-14-03123-f001:**
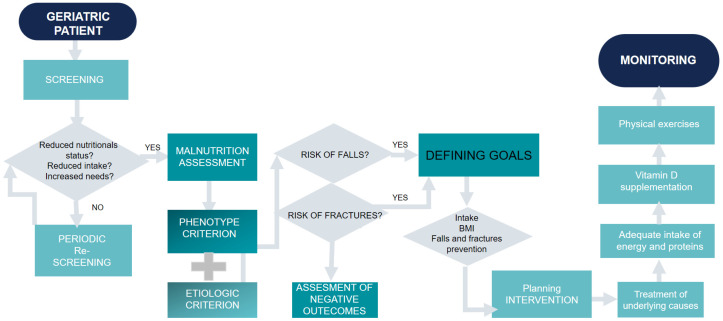
A summary of recommendations for malnutrition, fall, and fracture prevention.

**Table 1 nutrients-14-03123-t001:** Proposal of GLIM scheme and recommendations for adjustments for older adults.

*GLIM Scheme*	GLIM Recommendation	Geriatric Consideration
* **Risk screening** *	At risk of malnutrition Use validated screening tool	Chose the screening tool that includes other geriatric syndromes. Early detection of malnutrition predictors in males and females.
↓		
**Diagnostic assessment**	Assessment criteria Phenotypic: nonvolitional weight loss, low body mass index, reduced muscle mass Etiologic: reduced food intake or assimilation, disease burden/inflammatory condition	Necessity of continuing education in the field of anthropometric cutoff points for older adults (for professional and nonprofessional caregivers), verification of weight loss in patients with dementia or depression (collecting data from home caregiversas well), muscle strength, and function assessment. Necessity of pharmacological treatment assessment for reduced food intake.
*↓*		
**Diagnosis**	Meets criteria for malnutrition Requires at least 1 phenotypic and 1 etiologic criterion	Need of continuing education for professional caregivers, as well as home caregivers, to diagnose malnutrition, one of the geriatric syndromes.
↓		
**Severity grading**	Determine severity of malnutrition Severity determined based on phenotypic criterion	Contribution of other geriatric syndromes, assessment of its clinical outcomes, high risk of negative outcomes when coexisting.

↓—continue to the next step of GLIM scheme.

## Data Availability

Not applicable.

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
