# Peer review of "Malnutrition in Older Adults—Effect on Falls and Fractures: A Narrative Review"

_nutrients, 2022, doi:10.3390/nu14153123_

Round 1

Reviewer 1 Report

This review is interesting, and written well, so I recommend to publish this. 

Reviewer 2 Report

The authors have improved their review which is appreciated.

This manuscript is a resubmission of an earlier submission. The following is a list of the peer review reports and author responses from that submission.

Round 1

Reviewer 1 Report

It is a narrative review on a well-known topic and in a very general way.
Today systematic reviews are required on specific topics that provide us with scientific evidence or meta-analysis.

Author Response

Thank you for your recommendations and comments. I appreciate all your suggestions.

However, we don’t agree with the suggestion that it is necessary to include of meta-analysis to reach the aim of our article concerning malnutrition, falls and fractures. Each problem of geriatric syndromes such as malnutrition and mobility decline including falls and osteoporotic fractures appears to be well known . However, the subject concerning influence of malnutrition on falls and fracture is still under discussion. There is a gap in knowledge when considering the specific character of assessing risk of malnutrition, malnutrition and potential clinical outcomes that should be processed in clinical practice concerning falls and fractures. There is recently published  meta-analyses, narrative review and research concerning the general risk factors of vitamin D. However, there is a need to further study the connections of these factors. In clinical practice ,the main problem of geriatric population is not only the separate geriatric syndromes, but  the coexistence of factors mentioned above. The primary question seems to be the best practice of early detection of high-risk patients.

Our narrative review is a comprehensive, critical, and objective analysis of the current available knowledge of this topic.

The literature selected for this article is listed in the reference section and some af them are listed below.

Becker N, Hafner T, Pishnamaz M, Hildebrand F, Kobbe P. Patient-specific risk factors for adverse outcomes following geriatric proximal femur fractures. Eur J Trauma Emerg Surg. 2022 Apr;48(2):753-761; Besora-Moreno M, Llauradó E, Tarro L, Solà R. Social and Economic Factors and Malnutrition or the Risk of Malnutrition in the Elderly: A Systematic Review and Meta-Analysis of Observational Studies. Nutrients. 2020 Mar 11;12(3):737. doi: 10.3390/nu12030737.); Kong SH, Jang HN, Kim JH, Kim SW, Shin CS. Effect of Vitamin D Supplementation on Risk of Fractures and Falls According to Dosage and Interval: A Meta-Analysis. Endocrinol Metab (Seoul). 2022 Apr;37(2):344-358. Lee SH, Yu S. Effectiveness of multifactorial interventions in preventing falls among older adults in the community: A systematic review and meta-analysis. Int J Nurs Stud. 2020; doi: 10.1007/s43465-021-00478-3.; Reid IR, Bolland MJ. Sellier C. Malnutrition chez la personne âgée, dépister et prendre en charge [Malnutrition in the elderly, screening and treatment]. Soins Gerontol. 2018 Sep-Oct;23(133):12-17. French. doi: 10.1016/j.sger.2018.06.003.)

Reviewer 2 Report

The following points should be addressed in a revised version of this ms:

The authors should provide more insights into the age-related changes in body composition (BC). In their review they compile crude anthropometric variables with measures of detailed body composition as assessed by DXA or CT. It is worthwhile to mention that crude anthropometric variables like BMI (please note that the w/h-ratio is no longer used in the assessment of metabolic risks, it has been replaced by w only) have very limited value and may be used as a first selection criterion only. Furthermore, BMI may camouflage sarcopenia in overweight and obese patients, thus, this parameter may be misleading in the assessment of malnutrition in the elderly. 

It is recommended to clearly differentiate between the assessment of malnutrition risk and the diagnosis of malnutrition. These are two different issues.

The methods to assess BC are listed here. However the different methods are not 'equal', they have different outcomes and their data may add to each other. To get the idea of a detailed assessment of the nutritional state in an elderly population the authors are referred to the literature again (e.g., see J Gerontology A Bill Sci Med Sci 2016; 71:941-948; EJCN 2014; 68:1220-1227; EJCN 2017; 71:389-394).

It is unclear whether the so-called 'norms' fit to a population of older subjects. As mentioned before, the GLIM criteria refer to cut offs rather than norms.

In the context of this ms the value of laboratory data remains questionable. A screening of all the blood and plasma parameters mentioned here may not add much to the diagnosis of sarcopenia and frailty. In addition, the associations between specific deficiencies of micronutrients and the diagnostic criteria of malnutrition in the elderly is unclear. I feel that these are two different issues. First, the assessment of risks and the diagnosis of malnutrition according to the GLIM criteria. Second, to exclude specific deficiencies in micronutrients. The authors should also take some time to reconsider their list of laboratory parameters, there is need of some stratification according to the suspected micronutrient deficiency. Finally, it is worthwhile to reconsider whether the present cut offs of micronutrient deficiencies fit to a population of older subjects. At this point it may be fine to go back to an earlier para of this ms where the authors differentiated between subjects >65yrs and >75yrs. I doubt whether we presently have cut offs for different age groups.

The issue of age-related malnutrition should be strictly differentiated from the issues of post-operative risks of malnutrition. The pathophysiology of these two issues differs from each other. In an individual case they may add to each other.

The effects of interventions addressing malnutrition (or the risk of malnutrition) in an elderly population should be analyzed and described in more detail. A systematic review is welcome.

At the end the authors did not clearly differentiate between a nutritional intervention and a supplementation of Vita (or other micronutrients). While the former refers to nutrition the latter may be seen as 'nutrients as medicine'. Thus, the two interventions are not identical.

At the end the authors conclude that there is need of further studies. This may be true. However this conclusion is not very helpful unless the authors provide some more and detailed ideas about what kind of studies and what kind of outcome data are needed to improve our future  clinical practice.

Author Response

Thank you for your recommendations and detailed comments. I appreciate all your suggestions.

  1. We agree that crude anthropometric variables have very limited value, especially in older olds, but in clinical practice, they remain one of the most popular, easy measurable factors in the geriatric population. Moreover, some older research looked at programmes with anthropometric parameters that are not in currently in use. We emphatically agree that isolated measurements of anthropometric factors can be misleading, so we have highlighted the problem of sarcopenic obesity, however the detailed assessment of this problem was the topic of our other publication (in press, Polish Endocrynology).
  2. This has been underscored in the aim of the manuscript on page 2 (Diagnosis, epidemiology, estimation)
  3. Differences between methods and the recommendation for older adults were added to the manuscript
  4. We agree that the definition of “norm” is very difficult in older adults as very often it is rather so-called norm for young patients extrapolated to geriatric population. The term “norm” was changed to “cut off points” in the manuscript.
  5. An explanation as to why we presented laboratory tests in the section concerning malnutrition assessment has been added to the text. In our opinion, it is helpful element of comprehensive geriatric assessment, page 3 (Laboratory tests)
  6. We agree that post operative malnutrition is not the same as general malnutrition problems, but the typical of decompensation of elderly organisms are more often a malnutrition problem existing before the assessment of a post operative problem only. Cited literature addresses  urgent surgical interventions after hip fracture.
  7. There is a small amount of accessible data concerning influence of nutritional intervention on the combination of falls and fractures. However, we added the meta-analysis concerning the relationship between multifactorial interventions and falls.
  8. Thank you for your comments. We added the explanation of vitamin D supplementation and treatment vitamin D deficiency and conclusion concerning further studies.

Reviewer 3 Report

This review article is important. However, this will need to revise just a few points.

1) Please explain in more detail the purpose of this REVIEW. 2) Please create the main dissertation with TABLE.

3) Please summarize the main points. What are the limits of research? 4) Do you meet the regulations regarding how to cite documents?

Author Response

Thank you for your recommendations and comments. I appreciate all your suggestions. The purpose of this review was to make it clearer for the readers. The limitations of the cited research was also highlighted on page 7. The main points were summarized, presented graphically, and the limitations were also added and underscored. All the citations were carefully checked once more. Thank you for your suggestion to create a table for the main dissertation, however because of multifactorial relationship of malnutrition and falls and fractures it is very difficult to create a table that would be readable.

Round 2

Reviewer 1 Report

The authors have added basic concepts on nutritional assessment. The work continues without providing any new data or approach.

I still think that a narrative review is not high enough for a Nutrients level publication. First quartile journals only publish meta-analyses or at least systematic reviews.

Reviewer 2 Report

Sorry to say that the authors did not adequately address my concerns. The present review is still superficial. The authors mix up anthropometric and clinical chemical data and do not follow the present ESPEN recommendations.